# Effect of Mental Task on Sex Differences in Muscle Fatigability: A Review

**DOI:** 10.3390/ijerph192013621

**Published:** 2022-10-20

**Authors:** Patrik Vymyslický, Dagmar Pavlů, David Pánek

**Affiliations:** Faculty of Physical Education and Sport, Charles University, 16252 Prague, Czech Republic

**Keywords:** fatigue, muscle fatigue, sex differences, gender differences, mental task

## Abstract

Previous research demonstrated that there are observable sex differences in developing muscle fatigue when mental task during fatiguing activity is present; however, there is no available review on this matter. Therefore, this review aimed to summarize the findings of previous studies investigating the effect of mental task on muscle fatigue in men and women. To conduct the review, we utilized searches using the electronic databases Web of Science, PubMed, Scopus, and EBSCO Cinahl Ultimate. The studies included had no limited publication date and examined the effects of mental task on muscle fatigue in a healthy adult population of any age. The evaluation was performed using the following criteria: time to failure, or subjective scale in various modifications (visual analog scale—VAS, rate of perceived effort—RPE, rate of perceived fatigue—RPF, rate of perceived discomfort—RPD). A total of seven studies met the set criteria, which were subsequently analyzed. Heavy mental task (more demanding math tasks) can reduce the time to failure for both men and women, with the reduction being more pronounced for women than for men. For light mental task (simple math tasks), no reduction in time to failure was observed to a great extent. The mental task in any of the included studies did not affect the subjective perception of fatigue, effort, discomfort, or pain. Although the studies investigating the effect of mental task on sex differences in muscle fatigability are limited, based on our findings we can assume that in jobs requiring heavier mental task, women may be more prone to the faster development of muscle fatigue; thus, employers might consider paying attention to the possibility of adequate rest.

## 1. Introduction

Fatigue can be defined as a feeling of a lack of energy [1] and, if fatigue is not compensated, it accumulates, which can lead to overtraining, chronic fatigue syndrome (CFS) or immune disorders, and endanger human health [2]. Fatigue is most often distinguished as acute and chronic, central and peripheral, as well as mental fatigue and physical fatigue [3,4]. It is often stated that women have higher muscular endurance in some tasks (i.e., their muscles are more resistant to fatigue) than men [5,6,7,8,9,10]. 

According to previous studies, there is a relationship between fatigue (mental and physical) and musculoskeletal disorders [11,12]. Musculoskeletal disorders are common, especially in occupations that require static loads and repetitive movements—such as computer or assembly-line work [13]. Women generally report a higher prevalence of musculoskeletal disorders than men, but the explanation for these differences is not well understood [14]. More importantly, there is past and present bias in physiology research that favor studying more males than females with possible presumption about the absence of sex differences in fatigability [15].

Mental task appears to be an important factor in relation to fatigability [16,17] and, therefore, also to musculoskeletal problems [18,19]. It was previously observed that neck muscles show higher activity when a person is exposed to mental task [20,21]. In many daily activities, there is a need of performing mental tasks and motor performances simultaneously; thus, mental fatigue might influence physical and vice versa [22]. Performance declines might result from fatigue [23], but the effect of mental task on fatigue is not explored enough [22]. There is some evidence that the decline in physical performance might be a result of mental-fatigue-caused increased intracortical inhibition [24]. Mental fatigue might also lead to workplace injuries and accidents, thus impairing workplace safety [25]. Mental task is also a variable by which the sex differences in fatigability can be altered [26]. In response to mental task, sympathetic neural activity (especially heart rate) is elevated in women more so than in males [22]. In contrast to males, women’s elevated sympathetic responses may affect neuromuscular function by altering skeletal muscle blood flow and contractile performance [27]. Previous work in this field suggested that more research is needed to determine the severity of mental task’s influence on muscle fatigability and women’s susceptibility [15]. The aim of this review is to examine the impact of concurrent mental task on sex differences in muscle fatigability.

The findings of this work might aid in the identification of which sex may be more prone to injury when mental task is present and, thus, can contribute to the development of sex-specific guidelines with a goal to avoid not only work-related injuries and accidents. Furthermore, without considering sex as a variable in fatigability, rehabilitation, or training, we cannot be sure of providing sufficient recovery, rehabilitation, and training protocols. In addition, new research is emerging in the healthcare field, for example, in nutritional strategies [28] addressing the same topic discussed here, that is of importance to sex-specific research, which is currently insufficient. Investigating and understanding sex differences is tremendously important especially because of, as stated before, the current huge bias among (not only) physiology studies which, throughout the whole of research history, have favored studying more males than females, or did not pay attention to sex at all. 

## 2. Materials and Methods

### 2.1. Search Strategy and Information Sources

A search of publications was conducted in May 2022 in these databases: PubMed, Web of Science, EBSCO Cinahl Ultimate, and Scopus. Authors followed the PICO model by which the research question was set. The search strategy, including keyword combinations, is presented in Table 1. 

### 2.2. Eligibility Criteria

After duplicates removal, the main author and first co-author screened abstracts for eligibility. In case of disagreement, both reviewers debated until an agreement was reached. For the selection of results, the inclusion criteria were as follows: articles‘ publishing dates were not limited, participants in studies were not limited by age or muscle used during the fatiguing task, and the authors of the studies induced mental task on the probands before, during, or after the fatiguing task. Outcomes were as follows: time to failure, subjective scales (visual analog scale—VAS, rate of perceived effort—RPE, rate of perceived fatigue—RPF, rate of perceived discomfort—RPD). Studies were excluded if they had non-human participants or included participants with neurological or systemic diseases. 

### 2.3. Study Selection, Data Collection, and Summary Measures

After screening the titles and abstracts for the inclusion criteria, the selected abstracts were obtained in full texts. Full-text articles were selected independently by two reviewers and, in case of disagreement, the discussion between reviewers started until an agreement was reached.

A total of 430 studies were identified. After removing duplicates, the number decreased to 327 and, after screening the abstracts, 17 studies were compliant and subjected to in-depth investigation. After further investigation, 10 studies were discarded due to undesirable outcomes and a total of 7 were used for the final processing of the work. Authors followed the PRISMA model, and the screening process is summarized in the PRISMA diagram (Figure 1).

## 3. Results

### 3.1. Study Characteristics 

The following data from the included articles were analyzed: demographic information (title, authors, journal, and year), characteristics of the sample (age, inclusion and exclusion criteria, and number of participants), study-specific parameter (study type, measurement tool used, type of mental task used), and results obtained. Table 2 and Table 3 describe the studies’ characteristics, extracted data, and also a possible limitation of each study.

### 3.2. The Outcome Measures

#### 3.2.1. Time to Failure

Time to failure without mental task was shorter in men in one study [22], while in other studies there were no sex differences [18,22,30,33].

Light mental task (subtracting 1 from 50 or 100) reduced time to failure in one study, and only in women [31], whereas in other studies it had no effect [22,30,33].

Heavy mental task (subtracting 13 from any four-digit number) reduced the time to failure in both sexes in two studies [22,30]. However, a greater effect was observed in women (19.4% versus 9.5% and 27.3% and 8.6%, respectively). In one study, a reduced time to failure was observed in women only and not in men [31]. Heavy mental task had no effect on reducing the time to failure in only one study [33].

Three studies [18,21,32] used mental task that the authors did not distinguish between light and heavy (Stroop test, document tracking, n-back test, multiplication by three randomly generated one-digit or two-digit numbers or following instructions on the monitor); however, no statistically significant differences were observed between the sexes in any of these studies.

#### 3.2.2. Subjective Scale

The subjective scales (VAS, RPE, RPF, RPD) were assessed in four studies [21,22,32,33], but no statistically significant effect was observed either in men or women.

## 4. Discussion

This article aims to summarize the findings of published experiments evaluating the effect of mental task on the change in muscle fatigue in men and women, measured by the reduction of time to failure and by subjective scales. Based on the results, it can be stated that the greatest influence on the reduction of time to failure was heavy mental task (subtraction of the number 13 from any four-digit number) and, to a lesser extent, light mental task (subtraction of the number 1 from the number 50 or 100). A total of seven publications were included in this work [18,21,22,30,31,32,33], two of which [22,30] further consisted of two separate studies (see Table 1).

A total of five studies examined sex differences alone [18,22,30,31,33], and two studies examined women only [21,32]. A total of six papers evaluated time to failure [18,22,30,31,32,33] and a total of four papers evaluated the subjective scale (VAS, RPE, RPD or RPE) [21,22,32,33].

Fatigue protocols were most often determined as a percentage of the maximum voluntary contraction (MVC), at 15% [18], 20% [22,30,31], 30% [32,33], 35% [18], and 55% [18]. One study used 7.5 min of pipetting as a fatigue protocol [21]. Most studies that used electromyography chose elbow flexors as the study muscle group [22,30,31], and one study chose the anterior tibialis muscle [33], and one the deltoid muscle [18]; one study evaluated hand muscles [32]. In elbow flexors, the difference in initial muscle strength in men and women may be a reason for different muscle fatigability, due to higher sympathetic activation in weaker individuals and thus otherwise-modulated type I and II muscle fiber activity [15,34]. Initial strength has also been identified as a primary predictor of differential muscle fatigue during heavy mental tasks [30].

All included studies, except one [21], implemented a fatigue protocol to failure. At least two measurements were performed in each study, always on the same subjects, with one of the measurements always being the control one (no mental task). The remaining measurements were performed with the current mental task [18,21,22,30,31,33], or immediately thereafter [32]. During each fatigue protocol, due to changes in the excitability of the motor cortex, higher attention is required than at rest, and further addition of mental task will result in reduced time to failure [23,35].

Muscle fatigue can be defined as a reduction in the ability of a muscle to generate strength [36,37,38] and can originate at different levels of the motor pathway. This is a very common phenomenon which complicates, for example, athletic performance or, for some diagnoses, even normal daily activities (ADL). Muscle fatigue is determined by many factors—a person’s level of fitness, diet, gender, age, type of exercise, and motivation [39,40,41]. During the onset of muscle fatigue, many biological changes occur—for example, the concentration of metabolites increases, the speed of signal conduction through the nerve (CV—conduction velocity) changes, and the number of active motor units changes [42]. Tracking these changes is often used to determine the degree of muscle fatigue [43]. The processes that take place during muscle fatigue are divided into peripheral and central (neural), which is essentially synonymous with the traditional division of muscle fatigue into peripheral fatigue and central fatigue [1,44]. 

Central fatigue is caused by mechanisms “outside the muscle”, i.e., within the central nervous system (CNS), which has the task of generating sufficient central stimulus (neural drive) to perform the required action—the origin and transfer of information to the appropriate group motoneurons and the subsequent maintenance of muscle activity [39]. Central fatigue results from a reduction of neural drive which results in a reduction in the number of recruited motor units and a reduction in their so-called “firing” [43,45]. The reason for the reduction of the neural drive is not reliably explained. One theory suggests an increase in the ratio of serotonin to dopamine, which is associated with a subjective feeling of fatigue and lethargy, leads to a faster onset of muscle fatigue [46]. In the matter of women’s higher impact of mental task (cognitive stressor) on reduction of time to failure, we can presume that this might be a result of different active neural networks compared with men. As other authors stated, for example, anterior cingulate cortex activates differently in men and women but, overall, there is not enough evidence to sufficiently explain sex differences in the functionality of neural networks and pathways. [22].

Peripheral fatigue is caused by processes directly “in the muscle”. When examining peripheral fatigue, the goal is to determine which process involved in muscle function is the site of fatigue and what is the reason for the reduction in muscle strength. The primary site may be the excitation–contraction process, the neuromuscular junction, or the myofibrillar complex [39]. The myofibrillar complex includes regulatory proteins (troponin, tropomyosin) and contractile proteins (actin, myosin), while their proper cooperation is also necessary. A specific example of a disruption of the myofibrillar complex may be, for example, insufficient cross-bridges, which may result in a reduction in muscle strength [39,47]. Most studies focus on the study of peripheral factors of muscle fatigue [48]; despite claims that both central and peripheral factors play a role in muscle fatigue, their relationship is not fully understood [39]. Sex differences in substrate utilization during physical activity are known—women use (oxidize) fats better than sugars and amino acids, while men have the opposite. Men, therefore, have a higher glycolytic capacity than women, and women, on the contrary, have a higher oxidative capacity—this difference in the energy metabolism of the muscle is dependent on the proportional differences of type I and type II muscle fibers [49]. However, during simultaneous physical and mental task, increased sympathetic activity can affect the muscle fibers contraction–relaxation mechanism and, in men, who are often stronger and, thus, have proportionally more type II fibers, the selective effect of sympathetic activity on different fibers can cause a less rapid fatigability [30].

The percentage of slow and fast muscle fibers may not exactly differ between the sexes, but women have smaller type II fibers than men, so the whole muscle has a proportionally larger proportion of type I fibers—such a muscle is characterized by slower contraction, slower relaxation, and also greater resistance to fatigue [49]. This claim is supported by another study [50] where authors reported a more pronounced decrease in intramuscular pH during isometric contraction—that is, a higher acidity of the environment which surprisingly, in the same study, did not affect intersex muscle fatigue. Higher acidity is traditionally associated with higher concentrations of hydrogen cations (H+), which may contribute to the development of muscle fatigue [51].

Several subjective scales with different names were used in the studies we involved in this review. The scale of perceived exertion (RPE—rate of perceived exertion) or discomfort (RPD—rate of perceived discomfort) is a simple, indirect method capable of evaluating the degree of muscle fatigue in work, sports, or rehabilitation conditions. In most cases, this is a modification of Borg’s CR-10 scale [52]. A moderate to strong correlation was observed between the subjective ratings of effort (RPE) or discomfort (RPD) and objective indicators of muscle fatigue [53,54,55]. Micklewright [56] described the rate of perceived fatigue (RPF)—a validated questionnaire created by modifying Borg’s CR-10 scale. This scale takes values from 0 (no fatigue) to 10 (absolute fatigue and exhaustion—inability to continue). A strong correlation with objective indicators of muscle fatigue—blood lactate concentration, oxygen consumption, CO2 formation—was observed for this scale. The limitation of this work was the absence of correlation between RPF and the fatigue of individual muscles (measurable, e.g., with EMG), which was then added in another study [52] where a correlation was observed for these parameters as well.

Recording the electrical activity of the muscle using surface electromyography is a useful, non-invasive tool to objectify muscle fatigue [43]. The alteration of muscle activity that precedes muscle failure can be termed the myoelectric manifestation of fatigue [35,57]. The quality of the resulting EMG signal depends on many factors—e.g., the thickness of the skin under the electrodes, the location of the electrodes (whether they are close to tendons or motor points), or the so-called crosstalk, which is the interference of the electrical activity of the surrounding muscles and the monitored muscle. Using various techniques during the onset of muscle fatigue, specific parameters based on, for example, amplitude-based parameters, spectral analysis, time-frequency techniques, or non-linear parameters can be read from the electromyographic signal linear parameters [43].

One of the reasons for the differences in muscle fatigue in men and women may be a reduction in muscle blood flow, especially during isometric contraction, which may be more common in men than in women due to bigger muscle and greater strength, as there is more intramuscular pressure on arteries which may, thus, reduce muscle oxygen supply [58]. However, Fulco [59] questioned this claim because, with the same muscular strength, they found a higher rate of fatigue in men than in women. However, this research is again contradicted by the conclusion of Hunter [40], where they did not notice any differences in isometry between the strength-matched men and women. An earlier decrease in neural drive in men than in women may also play a role in different muscle fatigue, which may occur due to a higher accumulation of metabolites in active skeletal muscle and the subsequent inhibition of motoneurons by group III and IV sensory neurons [60].

Much of the current knowledge about sex differences in fatigue is based on monitoring static (isometric) contractions rather than dynamic (concentric or eccentric); although, in addition to the type of contraction, it also depends on contraction rate, intensity, and specific muscle groups involved [60]. For example, women reported lower fatigue rates in the slow concentric contraction of the elbow flexors, but not in the fast contraction [8,10]. Sex differences in fatigue have also been reported in sprinting—both cyclists and runners—always in favor of women [60,61,62], and also in long-distance running and long-distance cycling [9,63].

Within the individual phases of the menstrual cycle, endurance activities in the muscle show different sympathetic activity and substrate utilization; however, these fluctuations are minimal compared with the different fatigue of men and women [60]. On the other hand, a study by Salomoni [64] observed higher muscle endurance in isometric contraction in women than in men, but in women, at the end of the follicular and luteal phases (i.e. periods with significant estrogen depletion), these differences were smaller—i.e., they became tired earlier than women in other phases; however, they were still tired later than men.

Even though our study produced interesting results, we are aware of the limitations, which are especially the small number of studies that could be included in our study, which is due to the limited number of studies dealing with sex differences in muscle fatigue in combination with mental task. Furthermore, by the nature of the topic, experiments investigating sex differences cannot be Randomized Controlled Trials (RCTs); thus, they are not included in this review. Another limitation is the inhomogeneity of the mental task types used in individual experiments and, at the same time, the relatively small numbers of subjects included in the experiments. Future studies should focus on methodologically well-designed experiments (especially with a single type of mental task and more subjects) to help elucidate the increase in muscle fatigability under current cognitive task for different muscle groups and different fatiguing tasks and task conditions.

## 5. Conclusions

The results of this work suggest that mental task can reduce the time to failure in both men and women. This has been observed to a greater extent in heavy mental task using a more demanding mathematical task. In addition, women experienced a more significant reduction in time to failure than men. For light mental task using a simple mathematical task, a reduction in time to failure was observed to a minimal extent. The mental task in any of the included studies did not affect the subjective perception of fatigue, effort, discomfort, or pain. We can assume that in jobs requiring heavier mental task, women may be more prone to the faster development of muscle fatigue, and employers should pay attention to the possibility of adequate rest.

## Figures and Tables

**Figure 1 ijerph-19-13621-f001:**
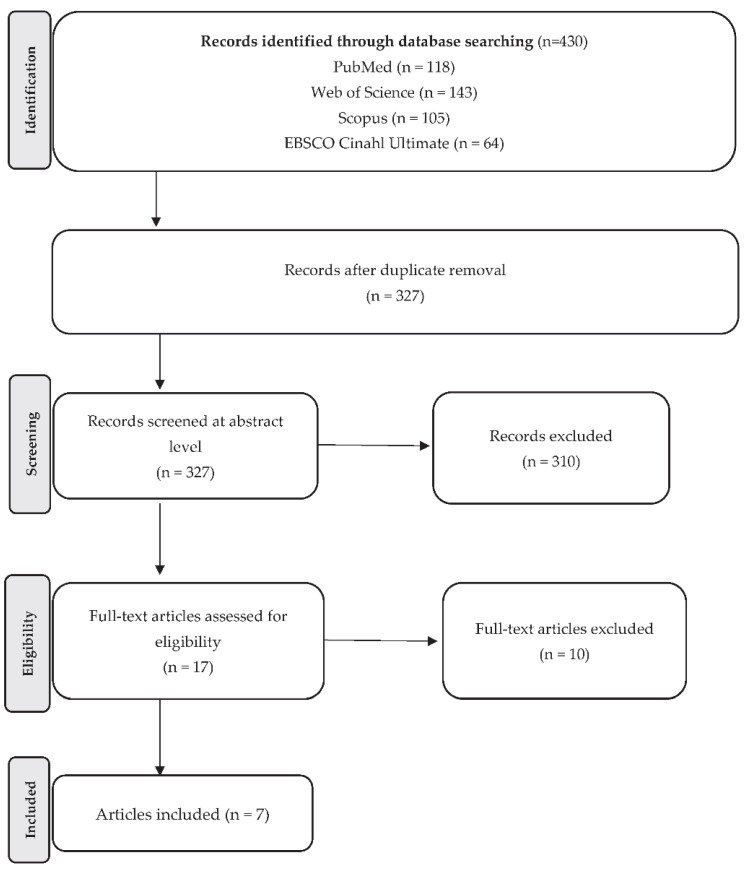
Flow diagram of the selection process according to Preferred Reporting Items for Systematic Review and Meta-Analysis (PRISMA) [29].

**Table 1 ijerph-19-13621-t001:** Keywords.

PICO	Keywords
P	(“sex differences” OR “gender differences” OR “sex difference” OR “sex variability” OR “gender difference” OR “gender variability”)
I	(“mental” OR “mental task” OR “cognitive” OR “cognitive task” OR “cognitive demand” OR “demand” OR “cognitive task” OR “task”)
C	NA
O	(“muscle fatigue” OR “muscle fatigability” OR “fatigability” OR “muscle fatiguing”

Abbreviations: NA: not applicable.

**Table 2 ijerph-19-13621-t002:** Studies included in the review.

Citation	Sample Size	Age (Years)	Movement Type Contraction Type Level of Effort Laterality	Study Type	Mental Task Type
Keller-Ross et al. (Study 1) (2014) [30]	Men = 26 Women = 29	Men = 20 ± 2 Women = 20 ± 3	Elbow flexion (90°) Isometrics 20% MVC to failure Left arm	Quasi-experiment	1. measurement: no mental task 2. measurement: heavy mental task 4 min before starting and during whole fatigue protocol (subtraction of number 13 from a four-digit number every 3 s)
Keller-Ross et al. (Study 2) (2014) [30]	Men = 14 Women = 9	Men = 20 ± 2 Women = 20 ± 4	Elbow flexion (90°) Isometrics to failure 20% MVC Left arm	Quasi-experiment	1. measurement: no mental task 2. measurement: light mental task 4 min before starting and during whole fatigue protocol (subtraction of number 1 from number 50 every 3 s)
Mehta et al. (2012) [18]	Men = 6 Women = 6	Men = 22.3 ± 1.86 Women = 21.3 ± 1.51	Shoulder abduction (90°) Intermittent isometrics (15 s contraction, 15 s rest) to failure 15%, 35% a 55% MVC Dominant arm (only right-handed)	Quasi-experiment	6 measurements in total: 15% MVC (control + mental task), 35% MVC (control + mental task), 55% MVC (control + mental task) mental task type = multiply by three randomly generated single-digit or double-digit numbers
Pereira et al. (2015) [31]	Men = 13 Women = 17	Men = 71 ± 5 Women = 70 ± 6	Elbow flexion (90°) Isometrics to failure 20% MVC Dominant arm	Quasi-experiment	1. measurement: no mental task 2. measurement: light mental (subtraction of number 1 from number 100 every 3 s) 3. measurement: heavy mental task (subtraction of number 13 from a four-digit number every 3 s)
Shortz et al. (2015) [32]	Women = 11	Women = 75.82 ± 7.4	Hangrip Intermittent isometrics (15 s contraction, 15 s rest) to failure 30% MVC Dominant arm (only right-handed)	Quasi-experiment	1. measurement: no mental task 2. measurement: after 1 hour long mental task (30 min Stroop test and 30 min n-back test)
Srinivasan et al. (2015) [21]	Women = 35	Women = 25 ± 5.8	7.5 min of pipetting Dominant arm	Quasi-experiment	1. measurement: no mental task 2. measurement: with mental task (watching on PC screen from where to pipette—in which row and column is the desired test-tube)
Vanden Noven et al. (2014) [33]	Men = 17 Women = 17	Men = 45.35 ± 3.45 Women = 43.85 ± 2.55	Ankle dorsiflexion (90°) Isometrics to failure 30% MVC Non-dominant leg	Quasi-experiment	1. measurement: no mental task 2. measurement: light mental task (4 min before and during fatigue protocol subtraction of number 1 from number 50 every 3 s) 3. measurement: heavy mental task (4 min before starting and during whole fatigue protocol (subtraction of number 13 from a four-digit number every 3 s)
Yoon et al. (Study 1) (2009) [22]	Men = 10 Women = 10	Men = 22 ± 4 Women = 22 ± 2	Elbow flexion (90°) Isometrics to failure 20% MVC Non-dominant arm	Quasi-experiment	1. measurement: no mental task 2. measurement: heavy mental task 4 min before starting and during whole fatigue protocol (subtraction of number 13 from a four-digit number every 3 s)
Yoon et al. (Study 2) (2009) [22]	Men = 11 Women = 8	Men = 20 ± 2 Women = 20 ± 4	Elbow flexion (90°) Isometrics to failure 20% MVC Non-dominant arm	Quasi-experiment	1. measurement: no mental task 2. measurement: light mental task (4 min before and during fatigue protocol subtraction of number 1 from number 50 every 3 s)

Abbreviations: MVC: maximum voluntary contraction.

**Table 3 ijerph-19-13621-t003:** Main findings and possible limitations of each study.

Citation	Main Findings	Possible Limitations
Keller-Ross et al. (Study 1/2) [30]	Time to failure: reduction in both sexes with sex differences (19.4% in women, 9.5% in men) Subjective scale: NA	No recordings of the day time when the measurement took place
Keller-Ross et al. (Study 2/2) [30]	Time to failure: no changes Subjective scale: NA	No recordings of the day time when the measurement took place
Mehta et al. [18]	Time to failure: reduction in both sexes without sex differences (most significant effect observed in 35% MVC) Subjective scale: NA	No recordings of the day of menstrual cycle during measurement No recordings of the day time when the measurement took place No assessment of baseline anxiety levels
Pereira et al. [31]	Time to failure: reduction in women, no changes in men Subjective scale: NA	Higher BMI in men (overweight) than in women (normal) No assessment of baseline anxiety levels
Shortz et al. [32]	Time to failure: no changes Subjective scale: no changes (RPD)	No recordings of the day time when the measurement took place Not included men Overweight and obese participants No assessment of baseline anxiety levels
Srinivasan et al. [21]	Time to failure: NA Subjective scale: no changes (RPF)	No recordings of the day of menstrual cycle during measurement No recordings of the day time when the measurement took place Not included men No assessment of baseline anxiety levels
Vanden Noven et al. [33]	Time to failure: no changes Subjective scale: no changes (RPE)	No recordings of the day of menstrual cycle during measurement No recordings of the day time when the measurement took place
Yoon et al. (Study 1/2) [22]	Time to failure: reduction in both sexes with sex differences (27.3% in women, 8.6% in men) Subjective scale: no changes (VAS)	No information about height and weight of the participants
Yoon et al. (Study 2/2) [22]	Time to failure: no changes Subjective scale: no changes (VAS)	No information about height and weight of the participants

Abbrevations: NA: not applicable; RPD: rate of perceived discomfort; RPF: rate of perceived fatigue; RPE: rate of perceived effort; VAS: visual analogue scale.

## Data Availability

Not applicable.

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
