# Peer review of "Effect of Mental Task on Sex Differences in Muscle Fatigability: A Review"

_ijerph, 2022, doi:10.3390/ijerph192013621_

Round 1

Reviewer 1 Report (New Reviewer)

Well done! 

It is not often I have the opportunity to review such a well organized and articulate manuscript. The subject is interesting and the presentation thoughtful. 

I look forward to seeing this information in publication. 

Author Response

Well done!

It is not often I have the opportunity to review such a well organized and articulate manuscript. The subject is interesting and the presentation thoughtful.

I look forward to seeing this information in publication.

Thank you very much for the time you spent reading our manuscript and above all thank you for the evaluation, which we greatly appreciate.

Reviewer 2 Report (New Reviewer)

Thank you for the opportunity to review this paper. While the topic is interesting, I feel that the manuscript is not appropriate for publication as it currently stands.

The aims, scope and rationale are not clearly defined. On the one hand, you talk about fatigue in general (Line 29), but the studies clearly investigated local muscle fatigue. While the implications for occupational injuries are important, I think the writing should be centered around fundamental science question - underlying mechanisms of fatigue and their interaction with cognitive demands and sex. 

In the discussion section, you outline quite a lot of fatigue mechanisms - Lines 198-252, but you should put that into context of sex differences. You do this later for some aspects (e.g. line 280-297 - this is what I would expect throughout the discussion). In my opinion discussion should be reorganized, perhaps by possible underlying mechanisms (central, peripheral, psychological), and for each, discuss why sexes differ. 

Although authors mention that differences in mental task type across studies, there is no attempt to compare studies in this view and trying to elucidate if sex differences emerge in specific combinations of motor and mental task. This is especially important as many studies do not support the sex differences or even no effect of mental task at all.

In the end, I think the paper would function better as a narrative review, where fatigue would be briefly introduced, and then one separate chapter would be used to compare and contrast the studies (in more depth than currently done in results), following by discussion of underlying mechanisms (with linking these to sex differences), concluding with a practical implications section. The methods section can still be kept, 

There are some minor grammar mistakes and the general writing style could be improved. 

Minor points (only as examples, a thorough proofread is needed)

- Line 10: perhaps "this review aimed", (to avoid saying study twice)

- Line 13-14:  needs to be completely rewritten

- Table 3, study 4: "no changes in men" instead of "in men no changes"

Author Response

Thank you for the opportunity to review this paper. While the topic is interesting, I feel that the manuscript is not appropriate for publication as it currently stands.

Thank you for your time that you devoted to our manuscript and for the recommendations for editing. Based on your comments and recommendations, we made the adjustments and believe that the quality of the manuscript has now improved.

  • The aims, scope and rationale are not clearly defined. On the one hand, you talk about fatigue in general (Line 29), but the studies clearly investigated local muscle fatigue. While the implications for occupational injuries are important, I think the writing should be centered around fundamental science question - underlying mechanisms of fatigue and their interaction with cognitive demands and sex.
  • In the discussion section, you outline quite a lot of fatigue mechanisms - Lines 198-252, but you should put that into context of sex differences. You do this later for some aspects (e.g. line 280-297 - this is what I would expect throughout the discussion). In my opinion discussion should be reorganized, perhaps by possible underlying mechanisms (central, peripheral, psychological), and for each, discuss why sexes differ.
  • Although authors mention that differences in mental task type across studies, there is no attempt to compare studies in this view and trying to elucidate if sex differences emerge in specific combinations of motor and mental task. This is especially important as many studies do not support the sex differences or even no effect of mental task at all.

We tried to incorporate all 3 comments. However, it is necessary to state that the strict separation of "types of fatigue" is difficult, because "local muscle fatigue" then leads to mental fatigue, and conversely, general fatigue is followed by musculoskeletal problems. In addition, if we look at the available studies, the nomenclature on this topic is not transparent here either. We have added a passage about central fatigue with relation to sex differences to the discussion. In connection with this, the resources were adjusted and renumbered.

In the end, I think the paper would function better as a narrative review, where fatigue would be briefly introduced, and then one separate chapter would be used to compare and contrast the studies (in more depth than currently done in results), following by discussion of underlying mechanisms (with linking these to sex differences), concluding with a practical implications section. The methods section can still be kept.

Thank you for the recommendation, but we allow ourselves to keep the nature of the research as it is. In connection with this, the introduction was added.

There are some minor grammar mistakes and the general writing style could be improved.

Minor points (only as examples, a thorough proofread is needed)

- Line 10: perhaps "this review aimed", (to avoid saying study twice)

- Line 13-14:  needs to be completely rewritten

- Table 3, study 4: "no changes in men" instead of "in men no changes"

Thank you for pointing out the grammar mistakes, all the mistakes you mention have been corrected.

Round 2

Reviewer 2 Report (New Reviewer)

Thank you for providing your response. I think the paper has somewhat improved, though I still think the problem could be approached from a different angle, especially in discussion. My final decision is to give "Accept" in the system but I rate the paper as of moderate quality. I will leave it up to the editors to make the final decision. 

This manuscript is a resubmission of an earlier submission. The following is a list of the peer review reports and author responses from that submission.

Round 1

Reviewer 1 Report

The research background of this paper is insufficient, and it does not fully explain the significance of the research. The analysis of research methods is not very scientific, and the discussion part cannot be directly related to the research data, and the depth and breadth of the discussion are not sufficient. In particular, it cannot support part of the conclusion, that is, women may be more prone to faster muscle fatigue. On the whole, this is not a very good research, which requires researchers to continue in-depth analysis and discussion. It is not suitable for publication at this time.

Reviewer 2 Report

The manuscript is original and of interest. I have only some comments to raise.

- Authors should clearly define the research question of the review.

- Authors mentioned the PICOS model (Participants, Interventions, Comparisons and Outcomes) however, it is not clear whether they used the PRISMA (Preferred Reporting Items for Systematic Reviews andMeta-Analyses). Moreover, PICO and PICOS is different. Which one they used? It depends on the study design considered for the review. Please edit table 1 according the abovementioned comment.

- In the paragraph 2.3, as well as in the flow diagram, authors declared that a total of 428 studies were identified. However, in the flow diagram the sum of the studies of the different databases is 430. Please double check it.

- Among the limitations of the study, authors should mentioned that for the review they did not consider only RCT.

Reviewer 3 Report

First of all, congratulations to the researchers.

The research article titled ''Effect of mental task on sex differences in muscle fatigability'' has a very high quality design.

The introductory part is adequately explained. The method section is pretty clear. The analyzes are correct. The data is well discussed in the discussion.

The research design and writing are impeccable.

I think it offers important information for practical practitioners. Introductory section should be improved. In the introduction, you should provide the readers with more detailed information on the subject.

I think it is ready to be published after the approval of the editors and reviewers.

Analysis seems appropriate.

Congratulations to all authors